# Self-Supported 3D PtPdCu Nanowires Networks for Superior Glucose Electro-Oxidation Performance

**DOI:** 10.3390/molecules28155834

**Published:** 2023-08-02

**Authors:** Kaili Wang, Shuang He, Bowen Zhang, Zhen Cao, Tingting Zhou, Jia He, Ganghui Chu

**Affiliations:** 1Laboratory of Xinjiang Native Medicinal and Edible Plant Resources Chemistry, Kashi University, Kashi 844008, China; wangkaili920823@163.com (K.W.); cjzbowen@126.com (B.Z.); 2College Chemistry & Chemistry Engineering, Weifang University, Weifang 261061, China; cczz250@163.com (Z.C.); zhoutingting1986@163.com (T.Z.); 3School of Materials Science and Engineering, Tianjin University of Technology, Tianjin 300384, China; 4Department of Nephrology, First Teaching Hospital of Tianjin University of Traditional Chinese Medicine, Tianjin 300193, China; 15620970867@163.com

**Keywords:** ultrafine PtPdCu nanowire, enzyme-free glucose oxidation, density functional theory

## Abstract

The development of non-enzymatic and highly active electrocatalysts for glucose oxidation with excellent durability for blood glucose sensors has aroused widespread concern. In this work, we report a fast, simple, and low-cost NaBH_4_ reduction method for preparing ultrafine ternary PtPdCu alloy nanowires (NWs) with a 3D network nanostructure. The PtPdCu NWs catalyst presents significant efficiency for glucose oxidation-reduction (GOR), reaching an oxidative peak-specific activity of 0.69 mA/cm^2^, 2.6 times that of the Pt/C catalyst (0.27 mA/cm^2^). Further reaction mechanism investigations show that the NWs have better conductivity and smaller electron transfer resistance. Density functional theory (DFT) calculations reveal that the alloying effect of PtPdCu could effectively enhance the adsorption energy of glucose and reduce the activation energy of GOR. The obtained NWs also show excellent stability over 3600 s through a chronoamperometry test. These self-supported ultrafine PtPdCu NWs with 3D networks provide a new functional material for building blood glucose sensors and direct glucose fuel cells.

## 1. Introduction

Diabetes is one of the common chronic diseases that seriously threaten human health. Real-time monitoring of blood glucose is helpful for patients and doctors to evaluate the condition, formulate reasonable and scientific treatment plans or adjust the existing plans. Glucose sensors can be divided into two categories, enzyme and enzyme-free glucose sensors. The important component of the enzyme glucose sensor is the enzyme electrode composed of glucose oxidase fixed on the bioactive interface. An enzymatic glucose sensor has high selectivity, sensitivity, and reversibility, but it also has problems that cannot be ignored [1,2]. One of the most common and difficult problems is the instability of the electrode caused by the properties of the enzyme. The use of glucose oxidase is limited to environments with pH 2–8, temperatures less than 40 °C, and ambient humidity conditions. An enzyme sensor is complicated, reproducible, and is greatly interfered with by oxygen. Faced with the increasing number of diabetic patients, the high cost of production and short service life of enzyme sensors seem to be difficult to meet the demand for wide application. Therefore, researchers have turned their attention to the enzyme-free glucose sensor. Enzyme-free glucose sensor utilizes a solid catalyst to oxidize glucose at a certain potential. Compared with the enzyme glucose sensor, the enzyme-free glucose sensor is unaffected by enzyme denaturation and deactivation, leading to broader storage conditions and longer service life. Moreover, the cost of an enzyme-free glucose sensor is lower than that of an enzyme-glucose sensor. Finally, the stability and reproducibility of the enzyme-free glucose sensor are better than that of the enzyme glucose sensor. Nowadays, there are many methods for preparing enzyme-free glucose sensors, and the used electrodes include carbon materials, metal alloy, etc. [3,4].

Due to the limitation of electrode miniaturization, the active area of the glucose sensor is greatly reduced, and the active site is insufficient. New materials with excellent properties developed in the laboratory often cannot be applied because there is no space to fix them. To solve the problem of insufficient active sites, an important method is to prepare a three-dimensional network structure on the surface of the microelectrode to increase the specific surface area of the active surface of the electrode, including, for example, doping zero-dimensional quantum dots, metal nanoparticles, one-dimensional nanotubes, or two-dimensional graphene nanosheets in the electrode surface modifier [5,6]. Although these methods, the preparation is complex and requires harsh processing conditions and technology. The simplest network structure can be obtained by directly preparing electrodes with three-dimensional surface films, such as mesoporous carbon film electrodes, porous silicon film electrodes, porous metal film electrodes, etc. [7]. However, the three-dimensional structure of the electrode surface is limited by thickness. Increasing the thickness will increase the probability of coating shedding, which will affect the durability and stability of the sensor electrode. Therefore, it is necessary to find a solid coating that does not easily fall off, a simple and reliable preparation method for three-dimensional structure electrodes.

Hierarchical nanoporous films built from nanowires as nano-building blocks could inherit the properties of the nanowires. It not only has the conductivity and ductility characteristics of metal but also has the characteristics of small size and surface effects. In addition, its porous structure also makes it with large specific surface areas, low density, good permeability, high electrical conductivity, 3D interconnected networks, and monolithic architecture characteristics. Currently, it has been widely used in the fields of catalysis, sensing, separation, filtration, biological materials, fuel cells, and so on [2,8]. Therefore, nanoporous metal films are expected to play an important role in the enzyme-free glucose sensor. On the other hand, a bimetallic or multimetallic system will be a promising electrocatalyst because of its ideal electronic and catalytic properties. Due to the synergistic action of the two materials, the bimetallic system can significantly increase the electro-oxidation of glucose and reduce the influence of interference and toxic substances on the electrode. In addition, most bimetallic systems have received much attention due to their ability to operate in endosomatic pH environment, for instance, PtPb, PtRu, PtNi, PtPd, PtAu, AuCu, NiCu, etc., which have good catalytic performance for the electro-oxidation of small molecular [9,10,11,12,13,14,15,16,17,18]. However, the enzyme-free glucose sensor still faces some key problems, such as low sensitivity and unsatisfactory stability. Studies have shown that Pt is widely studied as an efficient electrocatalytic catalyst; however, pure Pt has strong chemical adsorbability and poor selectivity to organic compounds in physiological solutions [19]. Palladium is a metal known for catalyzing the oxidation of carbon monoxide and other hydrocarbons. In PtPd catalysts, Pd affects and modifies the surface environment and reactivity of Pt through geometric effects [20]. Copper, a highly oxygenophilic metal, has good water activation properties and can remove toxic intermediates such as CO. 

In this work, multimetallic hierarchical nanoporous films were constructed by ternary PtPdCu NWs in cooperation with adjustable atomic components using a brief surfactant-free approach. The resultant Pt_27_Pd_47_Cu_26_ NWs show excellent specific activities of 0.69 mA/cm^2^ for GOR, which are 2.6-fold more than that of Pt/C. The nanoporous PtPdCu film exhibits excellent stability over 3600 s by chronoamperometry test. We attribute the excellent glucose oxidation activity and outstanding stability to the optimized electronic structure of Pt and Pd for the charge transfer and surface strain by alloying Cu atoms, as a result of the optimized adsorption and desorption of glucose molecules. The network-like-based nanowire nanostructures of the PtPdCu NWs nanoporous film electrode result in large ECSAs of 41.6 m^2^/g and fast charge and mass transfer, as measured by EIS. To further understand the reaction mechanism of GOR on PtPdCu NWs nanoporous film catalysts, the density functional theory is also employed. Results indicate that GOR on PtPdCu NWs is a diffusion-controlled irreversible process. The PtPdCu ternary alloys have more electron exchange than pure noble metal Pt, providing a greater possibility for the subsequent reaction. The open ring energy barrier of glucose of the ternary alloys is lower than that of Pt_111_, proving that the GOR ability of the ternary alloy catalyst is superior to that of pure metal Pt.

## 2. Results and Discussion

The connected PtPdCu NW catalysts were prepared via a simple NaBH_4_-assistant reduction method, followed by a freeze-drying procedure. Transmission electron microscopy (TEM) images were employed to study the morphology of the PtPdCu NWs. Appendix A displays the refined structures of the PtPdCu NWs, constructed by ultrathin interlinked nanowires. The 3D interconnected NW networks presented an average diameter distribution of about 5 nm without any byproducts. X-ray diffraction (XRD) characterization was applied to analyze the intrinsic structure of the PtPdCu NWs sample, as presented in Appendix A. It demonstrated on different peaks (111), (200), (220) and (311) with characteristic structures, and a positive shift with respect to the peaks of the face-centered cubic (fcc) crystal structure, suggesting the formation of a Pt-Pd-Cu alloy, and the obvious characteristic peak at about 60° is the silicon wafer itself. The atomic ratio of Pt to Pd to Cu in the PtPdCu NWs catalysts determined by STEM-EDS spectrum (Appendix A) and ICP-OES was 27/47/26 and 29/48/23.3D PtPdCu NWs networks were then fabricated by the catalyst-coated membrane (CCM) method, which is facile to tune the electrode thickness (corresponding variational PtPd loading) by ultrasonic spray deposition, as illustrated in Figure 1a. The 3D PtPdCu NWs networks were sprayed to construct the self-supporting membrane electrode assembly (MEA), which contains a solid network of PtPdCu NWs with ~5 nm diameter held together via self-crosslinking. STEM-EDS elemental mapping images (Figure 1b) and line-scanning profile (Figure 1c) present a uniform distribution of Pt, Pd, and Cu elements throughout all NWs. Moreover, Figure 1(d_1_,d_2_) show there are two dominant interplanar distances on a single nanowire, 2.26 Å and 1.95 Å, attributable to the (111) and (200) plane of fcc PtPdCu alloy. SEM image of the exterior surface was utilized for the morphological characterization of 3D PtPdCu NWs networks (Figure 1e). They show 3D-network-like interconnected structures and are generated together with large amounts of open pores, forming an interconnected and self-supported “nano-micro” film structure. When the PtPd loading was 0.2 mg_PtPd_/cm^2^, the thickness of the electrode was about 450 nm, leading to a high-quality self-supporting electrode (Figure 1e).

The CV curves of the PtPdCu NWs, Pt/C, and Pd/C catalysts, scanned with 50 mV/s in 0.5 M KOH electrolytes, are used to analyze the electrochemical active surface area (ECSA) value, as shown in Figure 2a and Appendix A. In the forward scan of the Pt/C, Pd/C, and PtPdCu NWs catalysts, the oxidation peak of −0.7 V and −0.5 V vs. Hg/HgO was attributed to hydrogen uptake. And the peak appearing at 0.0 V and 0.4 V vs. Hg/HgO was put down to the formation of Pt-OH [21]. During the backward scan, there is a distinct cathodic peak between −0.28 V and −0.21 V vs. Hg/HgO, revealing that the reduction of oxidized Pt occurred in both catalysts. The result presents that PtPdCu NWs have a higher reduction peak current density (13.5 mA/cm^2^) and a significant upshift of cathodic peak potential of about 0.07 V in comparison with the Pt/C catalysts, indicating that the intermediate species, such as CO form GOR, was easily removed [22]. Based on the calculation result, Pt/C and Pd/C present a higher ECSA of 43.9 m^2^/g and 32.5 m^2^/g, which is consistent with the previous result [23,24,25,26], and PtPdCu NWs exhibit an ECSA value of 41.6 m^2^/g, revealing that the ECSA of PtPdCu NWs catalysts are slightly lower than that of Pt/C but higher than that of Pd/C.

CVs analysis was performed in 0.5 M KOH and 0.1 M glucose solution to test the GOR performance, as shown in Figure 2b. During the forward scan, the onset potential to GOR for the PtPdCu NWs (~−0.43 V vs. Hg/HgO) was more negative than that of the Pt/C catalysts (about −0.33 V vs. Hg/HgO), proving rapid oxidation kinetics for glucose. In the forward scan, peak 1 at ~−0.65 V vs. Hg/HgO is put down to the dehydrogenated adsorption of glucose for the Pt/C and PtPdCu NWs catalysts. Peaks 2 at ~−0.1 V vs. Hg/HgO is attributed to the further oxidation of the intermediates. The peaks of 3 at about 0.3 V vs. Hg/HgO could often be ascribed to the further oxidated to 2-keto-gluconate with the transfer of 4 electrons. The electrocatalytic activity of anodic peak current density toward GOR on the PtPdCu NWs is 290 mA/mg_PtPd_, 2.4-folds and 2.8-folds above that of Pt/C and Pd/C, respectively. For specific activities, the GOR peak current density of PtPdCu NWs (0.69 mA/cm^2^) was 2.6-folds than that of the Pt/C (0.27 mA/cm^2^), revealing that PtPdCu NWs can outperform commercial Pt/C in the matter of both mass and specific activity (Figure 2c). Appendix A illustrates the GOR on PtPdCu NWs in 0.5 M KOH electrolyte at different concentrations of glucose with 0.1 M, 0.05 M, and 0.025 M. The result shows that the peak current density increases gradually with the increase of glucose concentrations from 0.025 M to 0.1 M. The oxidizing current density decreased slightly at lower glucose concentrations. The results demonstrated that the self-supported ultrafine PtPdCu nanowires with 3D networks provide a new functional material for building blood glucose sensors. In summary, the alloying of Cu and Pd atoms into Pt significantly improves the electrocatalytic activity of GOR, which can be demonstrated by comparing the catalytic activity of PtPdCu NWs catalyst with those of previously reported catalysts, as presented in Appendix A.

Figure 2d presents typical LSV curves tested at 0.5 M KOH and 0.1 M glucose electrolyte with a scan rate of 1 mV/s, decreasing the mass transfer effect and shows a steady-state polarization curve for GOR on Pt/C and PtPdCu NWs, reflecting kinetic processes of GOR [27,28]. Regarding the LSV curves in Figure 2d, the oxidation current of the PtPdCu NWs presents higher than Pt/C on the whole potential range. As expected, this shows a very low onset potential at −0.6 V vs. Hg/HgO for PtPdCu NWs, which makes this catalyst outstanding for GOR. As shown in Appendix A, the CO stripping peaks of the PtPdCu NWs (~−0.34 V vs. Hg/HgO) located at a more negative position by 200 mV than that of Pt/C (~−0.14 V vs. Hg/HgO) [29], showing better CO oxidation reactivity due to the incorporation of Cu and Pd into Pt lattice. The XPS characterization in the work showed that the electronic structure of PtPdCu NWs after the stability test had a stronger oxidation capacity to CO. We have performed CO-stripping experiment for PtPdCu NWs before and after the stability test, revealing that in the catalytic cycles, PtPdCu NWs has efficient oxidizing ability of CO (Appendix A). Thus, the resulting CO will first cause further oxidation in the modulated catalyst. If the CO remains unoxidized by PtPdCu, and the catalytic cycle stops at the CO intermediate, the CO will gather to form bubbles and dissolve into the solution without poisoning catalyst active sites. Meanwhile, the ECSA calculated from CO stripping of Pt/C and PtPdCu NWs were 37.9 m^2^/g_Pt_ and 20.5 m^2^/g_PtPd_. The ECSA of Pt/C obtained by CO-stripping and hydrogen adsorption/desorption area are almost equal; however, the ECSA of PtPdCu NWs obtained by CO-stripping was lower than that by hydrogen adsorption/desorption area, revealing the PtPdCu NWs catalysts surface’s defectiveness increases more than commercial Pt/C [30]. Chronoamperometry (CA) tests were performed to analyze the stability of PtPdCu NWs and Pt/C. Figure 2e shows CA curves tested at −0.2 V (vs. Hg/HgO) for 3600 s in a 0.5 M KOH with 0.1 M glucose solution. The initial current density of PtPdCu NWs for GOR is up to 60 mA/mg. The rapid decline of current indicates that PtPdCu NWs and Pt/C catalysts may be attributed to the surface atoms rearrangement of catalysts, or adsorbates in glucose (e.g., CO-like intermediates) poisoning the catalyst surface, or a decrease in glucose concentration around the electrode. The as-prepared PtPdCu NWs catalyst presented excellent stability compared to the commercial Pt/C catalysts during operation. After CA tests, the morphology changes of PtPdCu NWs catalysts were characterized. As presented in Appendix A, the morphology of the NWs exhibits no significant changes after the stability test, suggesting good stability in an alkaline media. Electrochemical impedance spectroscopy (EIS) test was also performed at −0.2 V vs. Hg/HgO. These Nyquist plots can be fitted by an equivalent circuit, as shown in the inset of Figure 2f and Appendix A, where R_1_ is the bulk resistance of the electrochemical system, R_2_ is Faradic charge transfer resistance, C is the electrical double-layer capacitor, and W is the Warburg impedance. Moreover, the 3D self-supported PtPdCu NWs electrode presents a considerable value of R_1_ (~21.9 Ω) with a commercial Pt/C electrode of R_1_ (~20.9 Ω). However, a remarkably lower value of R_2_ (~1.92 kΩ) than that of Pt/C (~10.39 kΩ) indicates that PtPdCu NWs can facilitate electron transfer and substance diffusion. The PtPdCu NWs have better conductivity due to the lower electron transfer resistance, consistent with the best GOR performance [31,32]. Amperometric glucose detection was done at −0.1 V vs. Hg/HgO (Appendix A). The detection was performed in a 0.5 M KOH electrolyte with separate glucose, acetic acid, urea, and glucose. Appendix A presents the selective current response of PtPdCu NWs catalyst on adding 1 mM glucose, acetic acid, and urea, in turn. The results reveal that the PtPdCu NWs catalyst showed obvious selectivity for glucose only.

The Pt surface chemical state of PtPdCu NWs before and after the CA test was investigated by XPS. At first, as presented in Figure 3 and Appendix A, the peak of Pt^0^, Pd^0^, and Cu^0^ in PtPdCu alloy NWs are located in 71.9 eV, 336.2 eV, and 932.8 eV, respectively, which have negative shifts of 0.9 eV than that of pure Pt (71.0 eV), a positive shift of 1.1 eV for pure Pd (335.1 eV), and a positive shift of 0.2 eV for pure Cu (932.6 eV), indicating alloying Pd and Cu within Pt downshifted the Fermi level of Pt because of the strong electronic interactions involving Pt, Pd, and Cu [33,34]. For as-prepared PtPdCu NWs, the peaks of Pt^0^ 4f shifted negatively after the CA test. The negative shift of Pt 4f demonstrates a weak binding energy of CO produced by glucose oxidation on the catalyst surface owing to the charge transfer between Pt and Cu after the stability test. After the stability test, the Pt^2+^ 4f_5/2_ characteristic peak at 78 eV in XPS disappeared due to the surface Pt site adsorbing and activating glucose and intermediate species, and the electronic structure of the surface Pt changed, making the oxidation state Pt change to the metallic state Pt. The Pt/Pd/Cu molar ratios of PtPdCu NWs obtained from XPS are 1.93/3.98/1.81 and, revealing the formation of an evenly distributed PtPdCu alloy on the whole nanowires, which combines with the STEM-EDS spectrum. After the stability text, the Pt/Pd/Cu molar ratios of PtPdCu NWs is 1.82/3.67/1.46, occurring with Cu atoms dissolution during the GOR proceed.

Regarding the excellent ability of the PtPdCu NWs compared to that of Pt/C, further studies on the electrocatalytic behavior of the PtPdCu NWs were carried out. To understand the mass transport behavior of PtPdCu NWs and Pt/C electrodes, we further carried out CV measurements of the PtPdCu NWs and Pt/C electrode in 0.5 M KOH and 0.1 M glucose at different scan rates from 50 to 250 mV/s, and the respective profiles are displayed in Figure 4a,c. It was observed that the current response of the GOR increased with increasing scan rate, and further estimation showed that the current around −0.1 V versus Hg/HgO was proportional to the square root of the scan rate (*v*), obtaining a linear regression equation Ip = −6.19318 + 41.097*v*^1/2^, deriving a correlation coefficient (R_2_) of 0.996, which reveals that the GOR on PtPdCu NWs at sufficient potential is a diffusion-controlled process (Figure 4b) [26]. Moreover, the positive scan peak potential (Ep) increases with scan rates increasing. It shows the linear relationship between Ep and ln*v*, demonstrating the GOR on PtPdCu NWs is an irreversible process [28]. Moreover, the CV of PtPdCu NWs shows a negative shift at 250 mV/s, probably because of the gaseous CO or CO_2_ bubbles falling off the electrode surface. Then, glucose molecules are fully absorbed on the catalyst surface during CV scanning at different sweep speeds. Pt/C electrode shows a similar result as that shown in Figure 4, with a linear regression equation Ip = −0.41264 + 17.11839ʋ^1/2^, (R_2_ = 0.999), as presented in Figure 4d.

The alloying effect and surface modification of the catalyst can effectively reduce the activation energy of GOR by forming a synergistic effect among heterogeneous atoms while having the properties of each component metal. Adding transition metals to Pt electrodes as accelerators is considered one of the best methods to solve the stability of catalytic active agents [35,36,37,38]. The transition metal contains free d orbitals and unpaired d electrons, and the reactant molecules can form characteristic chemisorption with these free d orbitals, thus reducing the activation energy of complex reactions. To explore the electronic structure of glucose adsorption on the surface of ternary PtPdCu alloys, DFT simulations were conducted to calculate electron density and Mulliken charge on the surface of Pt111, Pt_1_Pd_1_Cu_2_-111, and Pt_1_Pd_2_Cu_1_-111, as shown in Figure 5, Appendix A. It can be seen from the different images that for the adsorption of the glucose ring and open ring, the O atom has strong electronegativity compared with the metal atom [39,40]. The electrons are transferred from the metal atoms, such as Pt and Pd, to the O atom, and the surface metal atoms have an electronic interaction conducive to the adsorption and subsequent reaction of glucose. In comparison with pure Pt111 (0.04 e), ternary alloys Pt_1_Pd_1_Cu_2_ (0.07 e) and Pt_1_Pd_2_Cu_1_ (0.06 e) of the Mulliken charge for glucose ring and open ring can be seen that ternary alloys have more than 0.2 electron exchange, providing greater reaction possibility for the subsequent reaction. Charge density redistribution is an important method to further study the interaction between chemical states and molecules on the catalyst’s surface.

The ring-opening process of cyclic glucose is the first step in the subsequent reaction. Here we compare the first step transition state search process of glucose on the plane of 111 with two ternary alloys and pure noble metal, as shown in Figure 6. It is found that the open ring energy barrier of glucose of the two ternary alloys is lower than that of Pt111, and the energy barrier of Pt_1_Pd_2_Cu_1_ is the lowest, which is consistent with the Mulliken charge conclusion, which proves that the electrocatalytic ability of the alloy is better than that of pure metal Pt. During GOR proceed, the mechanism on Pt atomic surface is as follows: at potentials below 0.3 V vs. RHE and in the double layer region between 0.3 and 0.6 V vs. RHE, the adsorbed dehydrated intermediates are further oxidized to form weakly adsorbed gluconates, and the adsorption intensity tends to weaken with the gradual increase in the applied potential. When scanned at potentials greater than 0.6 V, the adsorbed dehydrogenated intermediates were oxidized to form gluconolactone without cleavage of the C-O-C bond. Eventually, gluconolactone slowly desorbs and hydrolyzes to gluconic acid in the alkaline electrolyte, as presented in Appendix A [41]. Glucose oxidation can obtain a variety of high-value-added chemicals, including, for example, gluconic acid, gluconic acid, arabinose, formic acid, etc. In this work, glucose and its electro-oxidation products were qualitatively detected and analyzed by high-performance liquid chromatography–mass spectrometry. Appendix A presents the total ion current chromatogram of the electrolyte for GOR performed on the PtPdCu NWs catalyst. The response signal of gluconic acid (at 0.68 min) was the strongest. Therefore, the main product of GOR was gluconic acid, which corresponds to the mechanism of GOR in the alkaline solution.

To discuss the active site of the catalyst, we further studied the activity of different proportions of the catalysts. As-prepared Pt_1_Pd_1_Cu_2_ NWs were evaluated, as shown in Appendix A. The GOR current density of Pt_1_Pd_1_Cu_2_ NWs was 199.6 mA/mg_PtPd_, lower than that of Pt_1_Pd_2_Cu_1_ NWs. In Figure 6, it can be found that the open-ring energy barrier of two ternary alloys of glucose is lower than that of Pt111, and the energy barrier of Pt_1_Pd_2_Cu_1_ is the lowest. So, with the increment of Pd content in the alloy, the GOR performance of the sample would also improve. In addition, theoretical calculations show that compared to other metal sites, Pd has a more sufficient charge exchange with glucose molecules. In conclusion, apart from Pt, we can speculate that Pd may also be the active site of the GOR reaction [42,43]. Moreover, we have performed binary PtCu and PdCu NWs to reveal the role of components of Pt and Pd elements (Appendix A). PdCu NWs exhibit an excellent GOR activity of 189 mA/mg_PtPd_ compared to PtCu NWs (110 mA/mg_PtPd_), revealing that Pd atoms have high GOR performance.

## 3. Materials and Methods

### 3.1. Chemicals and Materials

Hexachloroplatinic acid hexahydrate (H_2_PtCl_6_·6H_2_O) and palladium chloride (PdCl_2_) were obtained from Aladdin. Glucose (C_6_H_12_O_6_, AR), copper chloride dihydrate (CuCl_2_·2H_2_O), hydrochloric acid (HCl, 37%), potassium hydroxide (KOH), and sodium borohydride (NaBH_4_, 98%, powder) were obtained from Sinopharm Chemical Reagent Co., Ltd., Shanghai. China. Commercial Pt/C (20%) catalysts were obtained from Alfa Aesar Co., Ltd., Ward Hill, MA, USA, and nafion solution (5 wt%) was obtained from Dupont. 

### 3.2. Synthesis of PtPdCu NWs Catalyst 

The PtPdCu NWs were prepared by a simple NaBH_4_ reduction method. At first, metal precursor solutions of H_2_PtCl_6_ (430 μL, 30 mM), H_2_PdCl_4_ (860 μL, 30 mM), and CuCl_2_ (430 μL, 30 mM) were added into 150 mL of ultrapure water and stirred at room temperature to make a homogeneous mixture. Then, freshly NaBH_4_ solution (3.5 mL, 50 mM) was rapidly added into the above mixed solution and stirred vigorously for 1 min to obtain a black suspension solution. After standing in a water bath at 45 °C for 6 hours, the product was washed several times with water and re-dispersed in water.

### 3.3. Materials Characterization

TEM images were obtained via a transmission electron microscope with a LaB_6_ Gun (TECNAI G2 Spirit TWIN, Hillsboro, OR, USA). HRTEM images were obtained by a high-resolution transmission electron microscope with FEG (Talos F200 X, Hillsboro, OR, USA). HRTEM-EDS mapping images were obtained by a high-resolution transmission electron microscope with FEG. X-ray diffraction (XRD, MiniFlex600, Rigaku, Japan) patterns were carried out on a Model D/max-rC X-ray diffractometer. 

### 3.4. Electrochemical Measurements 

The electrochemical analysis was carried out on an electrochemical workstation (CHI-760D) by using a standard three-electrode system. The catalyst ink was obtained by mixing 5 mg catalyst, 5 mL H_2_O, and ultrasonic for 1 hour. A certain amount of catalyst ink was coated onto the glassy carbon electrode (4.0 mm diameter) surface to form a nanoporous film with 25 μg_PtPd_/cm^2^ catalyst loading. Then, 4 μL Nafion (0.5 wt.%) was coated onto the catalyst film and we made sure the catalyst film was bonded firmly. The GOR tests were performed in an N_2_-saturated electrolyte by using the Hg/HgO as the reference electrode, and a carbon rod as the counter electrode. CV measurements were carried out on between −0.86 to 0.6 V vs. Hg/HgO in 0.5 M KOH with 50 mV/s or an electrolyte with 0.5 M KOH and 0.1 M C_6_H_12_O_6_ with 50~250 mV/s. LSV measurements were obtained on −0.86 to 0.6 V vs. Hg/HgO with 1 mV/s with 0.5 M KOH and 0.1 M C_6_H_12_O_6_ solution.

### 3.5. DFT Method

The Dmol_3_ module in the Materials Studio software package (version 7.0) was used to perform all calculations and optimizes the structural models of different systems. Perdew, Burke, and Ernzerhof (PBE) proposed a generalized gradient approximation (GGA) function composed of Becke exchange and related expressions. Double numerical positive polarization (DNP) is used as the base set. The convergence tolerance of energy is 2 × 10^−5^ Ha, and the maximum force and displacement of geometric optimization are 0.004 Ha Å^−1^ and 0.005 Å, respectively. The cut-off radius of the actual space global orbit is set to 4.5 Å. Brillouin District consists of 3 × 3 × 1 k point sampling. To avoid the interaction between periodic images, each vacuum space is set at least 15 Å in the z-direction. The transition state (TS) is calculated by the generalized LST/QST approximation.

## 4. Conclusions

In summary, novel ternary PtPdCu NWs were prepared by a simple NaBH_4_ reduction approach. The 3D PtPdCu NWs networks exhibited excellent cGOR performance in comparison with a Pt/C. This can be attributed to the synergetic effects of Pt with Pd and Cu atoms. Furthermore, the GOR kinetic of PtPdCu NWs catalyst was discussed. The results show that GOR on PtPdCu NWs is a diffusion-controlled irreversible process. In terms of glucose sensors, the cost of the self-supported PtPdCu nanowires studied in this work may limit its widespread use, but it will have more advantages in terms of environmental requirements [44]. In addition, the direct glucose fuel cell constructed with self-supported PtPdCu nanowires has significant cost and durability advantages compared with granular catalysts [45]. More importantly, compared to traditional granular enzyme-free glucose sensors, ultra-thin electrodes based on self-supporting nanowires have significant structural advantages. In general, the thickness of the ultra-thin catalytic layer of self-supporting nanowires can be controlled below 1 μm, and no additional ionic conductors need to be added to the catalytic layer [46]. Our innovative synthetic materials can synthesize a self-supporting ultra-thin, high-density membrane electrode catalytic layer. This will provide a possible electrode material for subsequent glucose monitoring applications and potential direct glucose fuel cells.

## Figures and Tables

**Figure 1 molecules-28-05834-f001:**
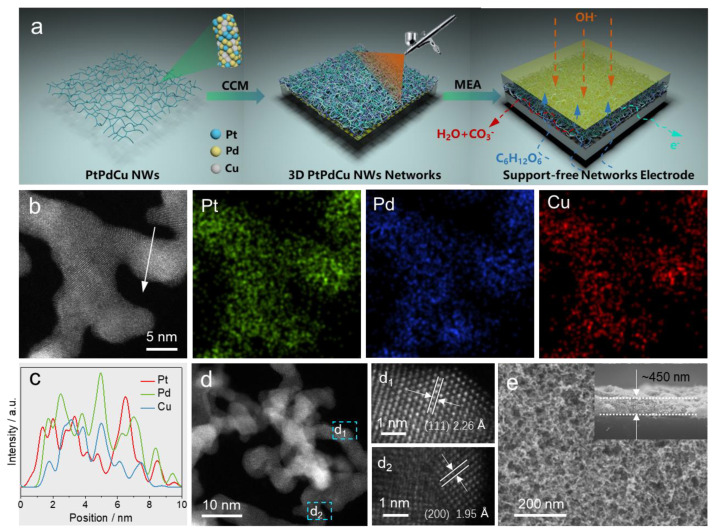
(**a**) Schematic illustration for fabricating 3D PtPdCu NWs networks electrode by spray method, the blue, yellow, and gray represent Pt, Pd, and Cu atoms, respectively. (**b**) HAADF-STEM and the corresponding EDS elemental mapping images of PtPdCu NWs. (**c**) STEM−EDS line-scanning profile form obtained from the location of the white arrow in (**b**). (**d**) HAADF−STEM images, (**d_1_**,**d_2_**) randomly selected magnified HAADF−STEM images. (**e**) Representative SEM images for 3D PtPdCu NWs networks electrode with 450 nm thickness and 0.2 mg_PtPd_/cm^2^ catalysts loading; insert in (**e**) shows the cross-sectional SEM image.

**Figure 2 molecules-28-05834-f002:**
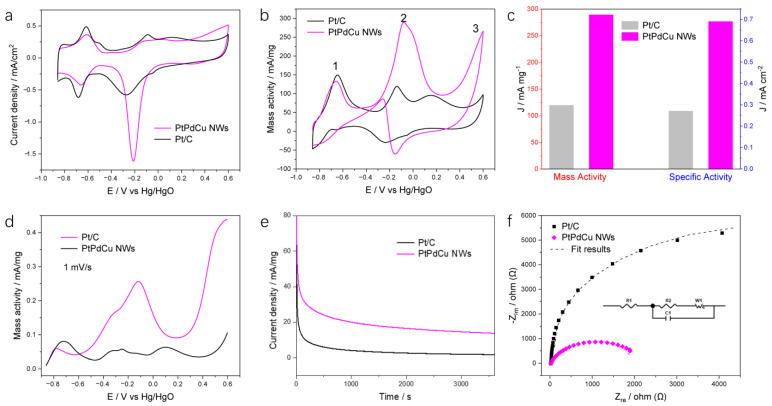
CV curves for the PtPdCu NWs and Pt/C catalysts tested in (**a**) 0.5 M KOH solution and (**b**) 0.5 M KOH containing 0.1 M glucose (50 mV/s scan rate). (**c**) Histograms of specific and mass activities of GOR. (**d**) LSV curves for PtPdCu NWs and Pt/C tested in 0.5 M KOH and 0.1 M glucose solution (10 mV/s scan rate). (**e**) Chronoamperometric curves at −0.2 V (vs. Hg/HgO) for 3600 s in a 0.5 M KOH with 0.1 M glucose solution. (**f**) Nyquist plots for PtPdCu NWs and Pt/C tested in 0.5 M KOH and 0.1 M glucose (a frequency range from 0.05 to 10^5^ Hz).

**Figure 3 molecules-28-05834-f003:**
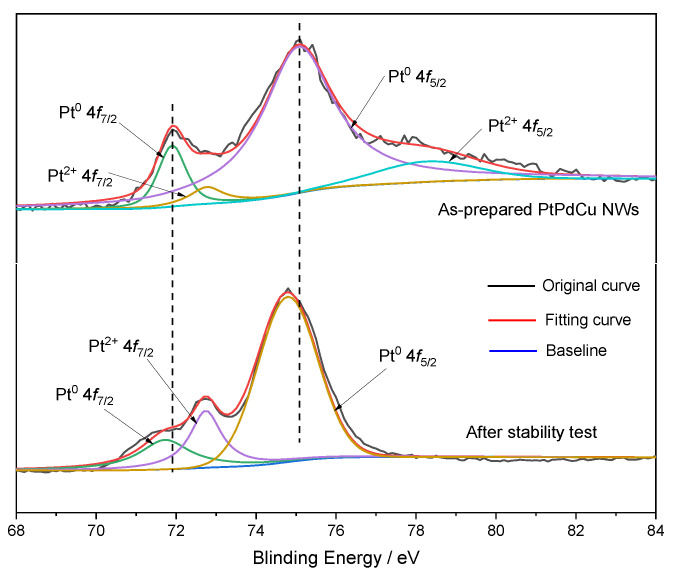
XPS patterns for Pt 4f region of the PtPdCu NWs before and after the stability test.

**Figure 4 molecules-28-05834-f004:**
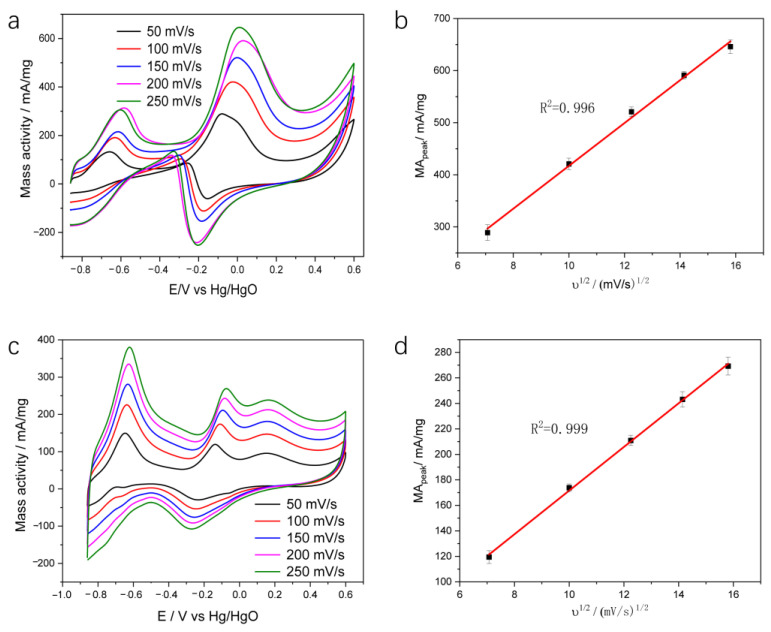
CV curves of the fabricated (**a**) PtPdCu NWs and (**c**) Pt/C in 0.5 M KOH and 0.1 M glucose with different scan rates from 50 mV/s to 250 mV/s; the dependence of anodic peak current of (**b**) PtPdCu NWs and (**d**) Pt/C.

**Figure 5 molecules-28-05834-f005:**
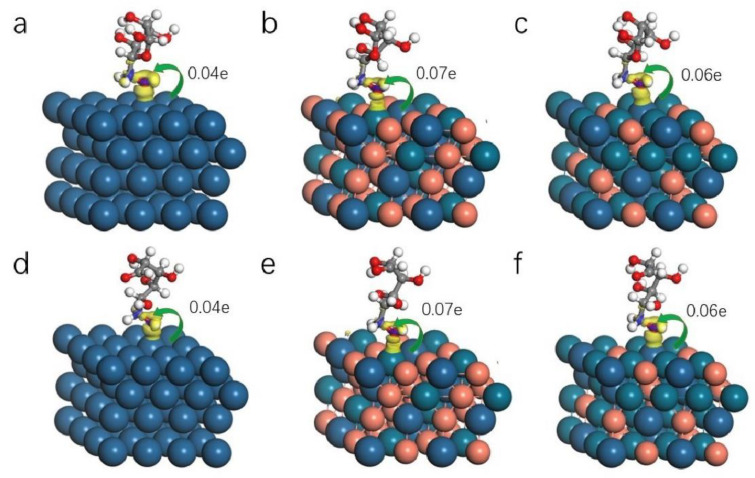
Electron density and Mulliken charge of Pt111, Pt_1_Pd_1_Cu_2_-111, and Pt_1_Pd_2_Cu_1_-111. (**a**–**c**) Glucose molecules adsorb on the three different models. (**d**–**f**) The figure represents opened-glucose molecule adsorb on the three different models. The blue area in the figure represents electron accumulation, and the yellow area represents electron loss. Each arrow represents the direction of electron transfer, and the number next to the arrow represents the relative transfer quantity. Blue and red balls represent Pt and Cu atoms, respectively, and the remaining balls are Pd atoms.

**Figure 6 molecules-28-05834-f006:**
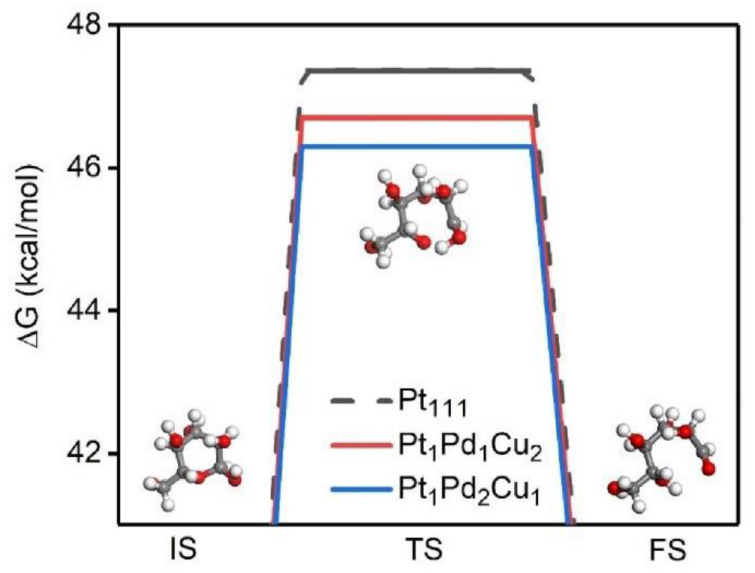
The transition state search of Pt111, Pt_1_Pd_1_Cu_2_-111, and Pt_1_Pd_2_Cu_1_-111 models between glucose and opened glucose.

## Data Availability

Not applicable.

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
