# Peer review of "Self-Supported 3D PtPdCu Nanowires Networks for Superior Glucose Electro-Oxidation Performance"

_molecules, 2023, doi:10.3390/molecules28155834_

Round 1

Reviewer 1 Report

In this manuscript, the authors have developed non-enzymatic and highly active electrocatalyst for glucose oxidation with excellent durability for blood glucose. The authors claim that it is a fast, simple, and low-cost NaBH4 reduction method for the preparation of ultrafine ternary PtPdCu alloy nanowires (NWs) with 3D network nanostructure. Although the manuscript has a lot of information and characterization, the authors fail to address the uniqueness of the work in terms of the process as well as the performance of the sensor. The manuscript it recommended for major revision.

11. Is the fabricated sensor marketable? If not, what are the prospects that need to be considered for it?

22. What is the unique feature of the sensor compared to the existing literature, and how does it stand out?

33. What are the drawbacks of the fabricated sensor?

44. The figure resolution should be improved for a better-quality manuscript.

55.  Calibration plot should be provided with error bars.

66. Does glucose detection fall in the rage of clinical detection?

77. Interference studies with relevant molecules should be performed.

88. The overall performance is because of the platinum, as shown throughout the manuscript. The role of Pd and Cu is completely neglected.

99.  “We can speculate that Pd may be the active site of GOR reaction” should be supported with literature.

110. Why did the authors use Nobel materials Pt and Pd? How would cost analysis be if one electrode is to be fabricated?

111. Interference studies must be performed with relevant molecules to increase the reliability of the sensor.

112.   Literature comparisons with similar types of materials should be tabulated.

113.   Important citations should be cited in the introduction. For e.g

a.       ACS Appl. Nano Mater. 2021, 4, 12, 13747–13760

b.       ACS Nano 2020, 14, 5, 5543–5552

c.       ACS Appl. Nano Mater. 2021, 4, 5, 4790–4799

d.       ACS Appl. Nano Mater. 2018, 1, 10, 5571–5580

e.       ACS Appl. Nano Mater. 2022, 5, 9, 13361–13372

The quality of English can be improved.

Reviewer 2 Report

The manuscript describes the preparation and characterization of PtPdCu catalysts for glucose electro-oxidation. This catalyst, which exhibits potent activity, is well characterized from mechanistic standpoints. I think that the results shown here would contribute to this research realm. I recommend publication of this work after the following points are addressed.

1)     In Introduction, the authors emphasize the superiority of inorganic PtPdCu catalysts in comparison with enzymatic glucose oxidation. While the inorganic PtPdCu catalysts exhibit higher durability and robustness than the enzymes, the selectivity of the inorganic catalysts are inferior to the enzymes. In application of this catalyst to practical analysis, the interference of other compounds must be minimized. The authors should address the weak points of this type of catalyst and show how to solve this problem.

2)     CV analyses: the authors state that the stronger binding ability of OH* is favorable for the oxidation of glucose. However, the principle of Sabatier indicates that stronger adsorption of intermediates slows the overall catalytic rate. The authors should show that the strengthen of the intermediate is not too low and not too high.

3)     The authors mentioned the further oxidation of glucose. However, no information of final product is given. The authors need to do product analysis to confirm their hypothesis.

4)     The authors state that CO, which is produced by glucose oxidation, affects the electronic state of Pt in XPS. However, if CO remains unoxidized by PtPdCu, the catalytic cycle stops at the CO intermediate. The authors should explain the role of CO in the catalytic cycles.

5)     The authors discuss the mechanism of the glucose oxidation and the role of components of PtPdCu. However, these hypotheses are not supported by experimental results.

Moderate editing of English language required

Round 2

Reviewer 1 Report

The manuscript can be accepted for publication.

Minor editing of the English language required

Reviewer 2 Report

The authors have addressed all the concerns that this reviewer pointed out. Now, the present form should be accepted.

It is OK, However, this reviewer thinks that English might be improved with advice of native speakers.